# Patterns of Presentation of Drug-Resistant Tuberculosis in Nigeria: A Retrospective File Review

**Olanrewaju Oladimeji** [1,2,*] , **Yasir Othman** [3] , **Kelechi Elizabeth Oladimeji** [1,4] , **Bamidele Paul Atiba** [2] , **Victor Abiola Adepoju** [5] and **Babatunde Adeniran Odugbemi** [6]

1 Department of Public Health, Faculty of Health Sciences, Walter Sisulu University, Mthatha 5099, Eastern Cape, South Africa
2 Faculty of Health Sciences, Durban University of Technology, Durban 4001, KwaZulu-Natal, South Africa
3 Department of Medicine, Hull University Teaching Hospitals NHS Trust, Hull HU3 2JZ, UK
4 Department of Laboratory Medicine and Pathology, Faculty of Health Sciences, Walter Sisulu University, Mthatha 5099, Eastern Cape, South Africa
5 Department of HIV and Infectious Diseases, Jhpiego (An Affiliate of John Hopkins University), Abuja 900271, Nigeria
6 Departments of Community Health & Primary Health Care, Lagos State University College of Medicine, Ikeja, Lagos 100001, Nigeria
* Correspondence: droladfb@gmail.com

**Abstract: Background:** An understanding of the patterns of drug-resistant tuberculosis (DR-TB) is needed to develop the best diagnostic tools and decide on optimal treatment combination therapies for the management of DR-TB in Nigeria. **Objective:** We aimed to investigate patterns of DR-TB for the five first-line anti-TB drugs over a period of seven years (2010–2016) and the associated clinical and socio-demographic factors. **Methods:** A retrospective study recruited 2555 DR-TB patients between 2010 and 2016 across the six geopolitical treatment zones in Nigeria. We determined DR-TB patterns based on standard case definition and their association with demographic and clinical information. Data were analyzed using Statistical Package for Social Sciences (SPSS) software. Independent predictors of DR-TB patterns/types were determined using bivariate and multivariate analyses with a statistical significance of $p < 0.05$ and a 95% confidence interval. **Results:** The majority of the participants were males, 66.93% (1710), 31–40 years old, 35.19% (899), previously treated, 77.10% (1909), had received at least two treatments, 411 (49.94%) and were multi-drug resistant, 61.41% (1165). The Southwest zone had the highest number of DR-TB cases, 36.92%. We found an upward trend in the prevalence of DR-TB from 2010 to 2016. Participants who had received one previous treatment showed statistically significant higher rifampicin resistance (59.68%), those with two previous treatments reported a statistically significant higher polydrug resistance (78.57%), and those with three or more previous treatments had a statistically significant higher multidrug resistance (19.83%) ($\chi^2 = 36.39$; $p = 0.001$). Mono-drug resistance and rifampicin resistance were statistically significantly higher in the southwest zone (29.48% and 34.12% respectively), polydrug resistance in the northcentral (20.69%) and south-south zones (20.69%), and multidrug resistance in the southwest (30.03%) and northcentral zones (19.18%) ($\chi^2 = 98.26$; $p = 0.001$). **Conclusions:** We present patterns of DR-TB across the six geopolitical zones in Nigeria. Clinicians should weigh in on these patterns while deciding on the best first-line drug combinations to optimize treatment outcomes for DR-TB patients. A national scale-up plan for DST services should focus on patients with previous multiple exposures to anti-TB treatments and on those in the Northeastern zone of the country.

**Keywords:** multi-drug resistant TB; mono-resistant; polyresistant; rifampicin-resistant

## 1. Introduction

Tuberculosis (TB) is a disease of public health importance and accounted for about 10 million new infections and 1.5 million deaths in 2018 [1]. Nigeria is one of the 30 countries

with the highest burden of TB, TB/HIV and drug-resistant TB (DR-TB) [2]. Nigeria ranks fourth in the world and first in Africa in the reported number of people with TB. The success of TB programs is measured by the effectiveness of TB treatments and the ability to prevent the emergence of DR-TB strains through early detection and prompt treatment with quality-assured drugs [1,3]. DR-TB is defined as a laboratory-confirmed resistance to one or more of the first-line TB medications [3]. Mono-resistant tuberculosis (MR-TB) shows resistance to one of the first-line anti-TB drugs; polyresistant TB (PDR-TB) is characterized by resistance to more than one of the first-line anti-TB drugs other than rifampicin and isoniazid; multi-drug-resistant tuberculosis (MDR-TB) exhibits resistance to at least both rifampicin and isoniazid [4,5]. Some of the factors leading to the development of DR-TB are lack of Directly Observed Therapy Short Course (DOTS), poor drug quality, interrupted TB drug supply, lack of standardized regimens in public and private facilities not engaged in the National Tuberculosis, Buruli Ulcer and Leprosy Control Program (NTBLCP), under- and over-prescription of fake anti-TB medications and low economic status [6,7]. Poor adherence to TB medications often leads to the emergence of DR-TB. In 2017, an estimated 558,000 people developed rifampicin-resistant TB, and of these, 82% had MDR-TB [1]. In 2018, DR-TB was estimated in 4.3% of new cases and 25% of previously treated cases in Nigeria [1]. Nigeria has DR-TB incidence, prevalence and death rate of 140/100,000, 200/100,000 and 20/100,000 population, respectively [3]. An accurate diagnosis of DR-TB is necessary for its successful treatment. The World Health Organization (WHO) has stated that only 16% of MDR-TB patients receive a correct diagnosis [8].

With shifting dynamics from a standardized to a more individualized TB treatment, information on drug resistance patterns at the population level is very crucial. Resistance to rifampicin, one of the most effective TB drugs, has been used as a surrogate marker for MDR-TB [9], since 85–90% of RR-TB is also resistant to isoniazid (INH) [10,11]. However, there are controversies over the appropriateness of using rifampicin resistance as a surrogate marker for MDR-TB. One study from Germany reported that 13.5% of patients with rifampicin-resistant TB did not have isoniazid resistance [12]. Recent studies also found that this surrogate marker was unreliable, as Xpert MTB/Rif did not accurately predict phenotypic MDR-TB in areas with low rifampicin resistance and low MDR-TB prevalence [13,14]. Using RR-TB as a proxy for MDR-TB may also increase over-diagnosis and over-treatment of DR-TB. Therefore, the need for an additional test such as Drug Susceptibility Test (DST) for isoniazid cannot be overemphasized [15]. Delayed laboratory results of the drug susceptibility test (DST) constitute a major problem and serve as a barrier to DR-TB treatment, as DST results take 2–3 months. A study from Adamawa State, Northern Nigeria, noted that the DST was not routine, several TB patients were assumed to have Drug-Sensitive TB (DS-TB), and only a few patients were referred for DR-TB [16]. Despite the limitations in result retrieval, DST still remains the best available laboratory test to determine the final treatment regimen for patients. However, it is recommended to carefully evaluate adding isoniazid to MDR-TB treatment in the context of Xpert-confirmed RR-TB when DST results are available [17].

Accurate information on DR-TB patterns is needed to inform on the use of different resistance diagnostic tools. There are limited data in Nigeria highlighting drug resistance patterns of DR-TB. This study, therefore, aimed to investigate patterns of DR-TB for the five first-line anti-TB drugs over a seven-year period (2010–2016) and the associated clinical and socio-demographic factors. Specific objectives were: (1) to determine the prevalence and distribution of DR-TB patterns in Nigeria between 2010 and 2016 and (2) to evaluate the socio-demographic and clinical factors associated with specific DR-TB patterns. The findings will inform diagnostic and treatment strategies for TB patients in countries like Nigeria with a high incidence of TB.

## 2. Methods

### 2.1. Study Design

This was a retrospective study that examined cross-sectional datasets of DR-TB patients managed in specialized DR-TB treatment facilities in the six geopolitical zones of Nigeria between 2010 and 2016.

### 2.2. Setting

#### 2.2.1. DR-TB Model in Nigeria

Nigeria is divided into six geopolitical zones with six states each and the Federal Capital Territory (FCT). The NTBLCP operates two models of DR-TB care, namely, the community ambulatory model and the hospital-based model for the Programmatic Management of DR-TB (PMDT) [2]. Nigeria started with the hospital-based model and later introduced the community model in 2013. Under the hospital model, DR-TB patients were hospitalized for the entire eight months of the intensive phase followed by 12 months of decentralized, ambulatory care for the continuation phase. In the community-based ambulatory model, patients are managed for the entire 20 months via ambulatory care in the community. The management in the continuation phase for both the community and the hospital model is the same. A treatment supporter daily visits the patient at home to administer the DOTS regimen, followed by biweekly visits of the patient to the DOTS center for drug pick up. The LGA TB supervisor also conducts a monthly home visit to the patient, while the State DR-TB team, a multidisciplinary team of experts, visits the patient quarterly.

#### 2.2.2. DRTB Case Finding and Diagnostic Coverage

Nigeria adopted the use of genexpert, which detects the TB bacteria in sputum samples, as the preferred method for diagnosing all forms of TB in 2015. Since then, the number of DR-TB cases rose by 35%, from 1686 in 2016 to 2286 in 2017, although this represented only 11% of the estimated MDR/RR-TB cases [2]. About 78% of DR-TB cases diagnosed in 2017 were enrolled in DR-TB care, but the situation later improved with the introduction of the DR-TB line listing tool by the NTBLCP. DR-TB activity implementation in Nigeria was funded by the Global Fund and the United States Agency for International Development (USAID) and implemented by the NTBLCP with the support of the Institute of Human Virology in Nigeria (IHVN) and the KNCV Tuberculosis Foundation.

In 2017, Nigeria had a total of 390 genexpert machines (an improvement from a total of 318 in 2016) for the diagnosis of *Mycobacterium tuberculosis* (MTB) and RR-TB cases. The number of laboratories capable of Line Probe Assay (LPA) evaluation for first-line TB drugs also increased from 5 in 2016 to 8 in 2017, while that of second-line drugs increased from 3 in 2016 to 7 in 2017 [2]. In 2017, 8 institutes provided culture and DST services, namely, the Nigeria Institute of Medical Research (NIMR, South West zone), the National Tuberculosis and Leprosy Training Center, Zaria (NTBLTC, North West); Zankli Hospital (FCT), Lawrence Henshaw Hospital (Cross Rivers, South South); Jos University Teaching Hospital (JUTH, North Central), the University College Hospital, Ibadan (UCH, South West); the University of Port Harcourt Teaching Hospital (UPTH, South South); and the Aminu Kano Teaching Hospital (AKTH, North West zone) [2]. There were no laboratories to assay both culture and DST in the South East and North-East in 2017, and DR-TB samples for these assays were sent for analysis to the closest geopolitical zones.

#### 2.2.3. DR-TB Treatment Coverage in Nigeria

The goal of the NTBLCP is to establish a minimum of one DR-TB treatment center in each geopolitical zone in Nigeria, but this was practically impossible until 2016–2017. In 2017, five national reference laboratories were upgraded to perform 2nd-line LPA; the use of a shorter DR-TB regimen was commenced, while program and facility staff were trained for the Programmatic Management of DR-TB (PMDT) and the shorter DR-TB regimen management [2]. Some of the challenges of DR-TB in Nigeria include an inadequate

Local Government Area (LGA) response to DR-TB service provision, delay in initiating therapies for diagnosed DR-TB patients and suboptimal and real-time data upload into the National Electronic TB Informational Management Systems (NETIMS) platform by LGA TB supervisors and the DR-TB treatment center staff.

### 2.3. Inclusion and Exclusion

A case had to have been notified between 2010 and 2016, tested for all four first-line anti-TB drugs (rifampicin, isoniazid, pyrazinamide, and ethambutol), and have available DST results in order to be included in the study. All age groups were represented. Due to insufficient data, information on second-line TB drug resistance was not included. All cases of any drug resistance, i.e., MDR-TB, PDR-TB, MR-TB, and RR-TB, according to the NTBLCP case definition, were also included.

### 2.4. Data Analysis

Extracted independent variables from the NTBLCP database included age, sex (male or female), HIV status (positive, negative and unknown), patient group (new or previously treated), current treatment zone address and DR-TB patterns. Descriptive analyses of DR-TB patterns (outcome variables) were initially performed and presented as case counts, and the proportion of each pattern was stratified by sex, age, patient group, number of previous treatment, treatment zone address and HIV status. Trends in each DR-TB pattern were analyzed using the chi-square test for linear trend and displayed using graphs. Comparison of case counts was conducted using the chi-square test with a significant *p*-value of less than 0.01. All data were analyzed using Statistical Package for Social Sciences (SPSS). Patient information was concealed at the beginning of the analysis to maintain privacy.

## 3. Results

### 3.1. Socio-Demographic and Clinical Characteristics of the Participants

As shown in Table 1, there was a total of 2555 study participants, including 1710 (66.93%) males and 845 (33.07%) females. The age distribution showed that the 31–40 years age group was the main one with 899 subjects (35.19%), while individuals who were 21–30 years old were 687 (26.89%) and those who were 41–50 years old were 465 (18.20%). The age group with 60-year-old and older subjects included 107 (4.19%) participants. In total, 77.10% (1909) of the participants had been previously treated, among which 411 (49.94%) received at least two treatments, while 567 (22.90%) were new cases.

**Table 1.** Frequency table of the study participants with bacteriologically confirmed DR-TB in Nigeria (2010–2016) (*n = 2555*).

| Variables | Frequency (*n*) | Percentage (%) |
|:---:|:---:|:---:|
| **Sex (*n = 2555*)** | | |
| Male | 1710 | 66.93 |
| Female | 845 | 33.07 |
| **Age (yrs.) (*n = 2555*)** | | |
| ≤20 | 201 | 7.87 |
| 21–30 | 687 | 26.89 |
| 31–40 | 899 | 35.19 |
| 41–50 | 465 | 18.20 |
| 51–60 | 196 | 7.67 |
| 60+ | 107 | 4.19 |
| **Patient group [#] (*n = 2476*)** | | |
| New | 567 | 22.90 |
| Previously treated | 1909 | 77.10 |
| **No of previous treatment [#] (*n = 823*)** | | |
| 1 | 302 | 36.70 |
| 2 | 411 | 49.94 |
| 3+ | 110 | 13.37 |

**Table 1.** *Cont.*

| Variables | Frequency (*n*) | Percentage (%) |
|---|---|---|
| **DR-TB category # (*n* = 1897)** | | |
| Mono-DR | 527 | 27.78 |
| RIF Resistant | 176 | 9.28 |
| Poly-DR | 29 | 1.53 |
| MDR | 1165 | 61.41 |
| **Zone current address # (*n* = 2438)** | | |
| NE | 221 | 9.06 |
| NC | 400 | 16.41 |
| NW | 368 | 15.09 |
| SS | 354 | 14.52 |
| SE | 195 | 8.00 |
| SW | 900 | 36.92 |
| **HIV Status # (*n*= 1303)** | | |
| Positive | 305 | 23.41 |
| Negative | 998 | 76.59 |

# Missing records.

About 61.41% (1165) of the respondents had multi-drug resistance, followed by 27.78% (527) with mono-drug resistance, and 9.28% (176) with rifampicin resistance. The Southwest zone recorded the highest number of DR-TB patients, corresponding to 36.92% (900), followed by the Northcentral zone, with 16.41% of DR-TB patients (400), and the Northwest zone, with 15.09% of DR-TB patients (368). The HIV-positive participants were 23.41% (305), whereas the HIV-negative patients were 76.59% (998), as shown in Table 1.

### 3.1.1. Pattern of Presentation of Bacteriologically Confirmed DR-TB Cases in Nigeria (2010–2016)

As shown in Table 2, the association between socio-demographic and clinical characteristics of the participants and the presentation of bacteriologically confirmed DR-TB was statistically significant in relation to the number of previous treatments and the current zone address. There was no statistically significant association with sex, age, patient group, and HIV status.

**Table 2.** Pattern of presentation of bacteriologically confirmed DR-TB cases in Nigeria (2010–2016) (*n* = 2555).

| | Mono-DR | | RIF-Resistant | | Poly DR | | MDR | | *df* | $\chi^2$ (*p-Value*) |
|---|---|---|---|---|---|---|---|---|---|---|
| | *n*(%) | 95% CI | *n*(%) | 95% CI | *n*(%) | 95% CI | *n*(%) | 95% CI | | |
| **Sex** | | | | | | | | | | |
| Male | 349 (66.22) | 62.08–70.13 | 120 (68.18) | 60.75–74.99 | 18 (62.07) | 42.26–79.31 | 801 (68.76) | 66.04–71.35 | 3 | 3.31 (0.511) |
| Female | 178 (33.78) | 29.87–37.92 | 56 (31.82) | 25.01–39.25 | 11 (37.93) | 20.69–57.74 | 364 (31.24) | 28.65–33.96 | | |
| **Age (yrs.)** | | | | | | | | | | |
| ≤20 | 36 (6.83) | 4.97–9.31 | 12 (6.82) | 3.57–11.61 | 2 (6.90) | 0.85–22.77 | 99 (8.50) | 7.03–10.24 | | |
| 21–30 | 161 (30.55) | 26.77–34.61 | 41 (23.30) | 17.27–30.25 | 7 (24.14) | 10.30–43.54 | 307 (26.35) | 23.90–28.96 | | |
| 31–40 | 177 (33.59) | 29.69–37.72 | 64 (36.36) | 29.26–43.94 | 12 (41.38) | 23.52–61.06 | 414 (35.54) | 32.84–38.33 | 15 | 14.80 (0.466) |
| 41–50 | 86 (16.32) | 13.41–19.72 | 33 (18.75) | 13.27–25.31 | 4 (13.79) | 3.89–31.66 | 214 (18.37) | 16.25–20.70 | | |
| 51–60 | 42 (7.97) | 5.95–10.60 | 18 (10.23) | 6.17–15.68 | 1 (3.45) | 0.09–17.76 | 92 (7.90) | 6.48–9.59 | | |
| 60+ | 25 (4.74) | 3.23–6.91 | 8 (4.55) | 1.98–8.76 | 3 (10.34) | 2.19–27.35 | 39 (3.35) | 2.46–4.54 | | |
| **Patient group** | | | | | | | | | | |
| New | 109 (20.76) | 17.51–24.44 | 46 (26.14) | 19.81–33.28 | 6 (20.69) | 7.99–39.72 | 253 (21.81) | 19.53–24.28 | 3 | 3.24 (0.366) |
| Previously treated | 416 (79.24) | 75.56–82.49 | 130 (73.86) | 66.72–80.19 | 23 (79.31) | 60.28–92.01 | 907 (78.19) | 75.72–80.47 | | |
| **No of previous treatment** | | | | | | | | | | |
| 1 | 80 (35.40) | 29.17–42.01 | 37 (59.68) | 46.45–71.95 | 2 (14.29) | 1.78–42.81 | 113 (32.47) | 27.77–37.56 | | |
| 2 | 118 (52.21) | 45.49–58.88 | 25 (40.32) | 28.05–53.55 | 11 (78.57) | 49.20–95.34 | 166 (47.70) | 42.51–52.95 | 6 | 36.39 (*0.001*) * |
| 3+ | 28 (12.39) | 8.39–17.41 | - | - | 1 (7.14) | 0.18–33.87 | 69 (19.83) | 15.98–24.34 | | |

**Table 2.** *Cont.*

| | Mono-DR | | RIF-Resistant | | Poly DR | | MDR | | df | $\chi^2$ (p-Value) |
|---|---|---|---|---|---|---|---|---|---|---|
| **Zone current address** | | | | | | | | | | |
| NE | 42 (8.09) | 6.04–10.76 | 11 (6.47) | 3.27–11.28 | 5 (17.24) | 5.85–35.77 | 114 (10.57) | 8.87–12.54 | | |
| NC | 74 (14.26) | 11.51–17.53 | 24 (14.12) | 9.26–20.27 | 6 (20.69) | 7.99–39.72 | 207 (19.18) | 16.95–21.64 | | |
| NW | 96 (18.50) | 15.39–22.06 | 53 (31.18) | 24.30–38.72 | 5 (17.24) | 5.85–35.77 | 193 (17.89) | 15.72–20.29 | 15 | 98.26 (0.001) * |
| SS | 96 (18.50) | 15.39–22.06 | 18 (10.59) | 6.40–16.22 | 6 (20.69) | 7.99–39.72 | 154 (14.27) | 12.31–16.49 | | |
| SE | 58 (11.18) | 8.74–14.18 | 6 (3.53) | 1.31–7.52 | 3 (10.34) | 2.19–27.35 | 87 (8.06) | 6.58–9.84 | | |
| SW | 153 (29.48) | 25.72–33.54 | 58 (34.12) | 1.31–41.77 | 4 (13.79) | 3.89–31.66 | 324 (30.03) | 27.37–32.83 | | |
| **HIV Status (*n*= 1303)** | | | | | | | | | | |
| Positive | 60 (23.26) | 18.24–28.90 | 27 (24.77) | 17.0–33.96 | 6 (42.86) | 17.66–71.14 | 120 (22.60) | 19.25–26.35 | 3 | 3.26 (0.353) |
| Negative | 198 (76.74) | 71.10–81.76 | 82 (75.23) | 66.04–83.0 | 8 (57.14) | 28.86–82.34 | 411 (77.40) | 73.65–80.75 | | |

* Statistically significant ($p < 0.05$); df = degree of freedom; $\chi^2$ = Chi-Square.

Participants with one previous treatment displayed a statistically significant higher rifampicin resistance (59.68%), while those with two previous treatments showed a statistically significant higher poly-drug resistance (78.57%), and those with three or more previous treatments had a statistically significant higher multidrug resistance (19.83%) ($\chi^2 = 36.39$; $p = 0.001$) (Table 2).

Mono-drug resistance was statistically significantly higher in the southwest (29.48%), rifampicin resistance in the southwest (34.12%), polydrug resistance, in the northcentral (20.69%) and south-south (20.69%) zones, and multidrug resistance, in the southwest (30.03%) and northcentral (19.18%) ($\chi^2 = 98.26$; $p = 0.001$) zones (Table 2).

3.1.2. Trend of the Pattern of Presentation of Laboratory-Confirmed DR-TB Cases by Year (2010–2016)

Figure 1 shows the trend of laboratory-confirmed drug-resistant TB cases between 2010 and 2016. There was an upward trend in the prevalence of mono-drug, rifampicin, polydrug and multi-drug resistance from 2010 to 2016, with rifampicin resistance having the highest prevalence (52.8%) in 2015 but showing a sharp drop thereafter. Figure 2 shows the map with embedded bar chart of the pattern of presentation by zone.

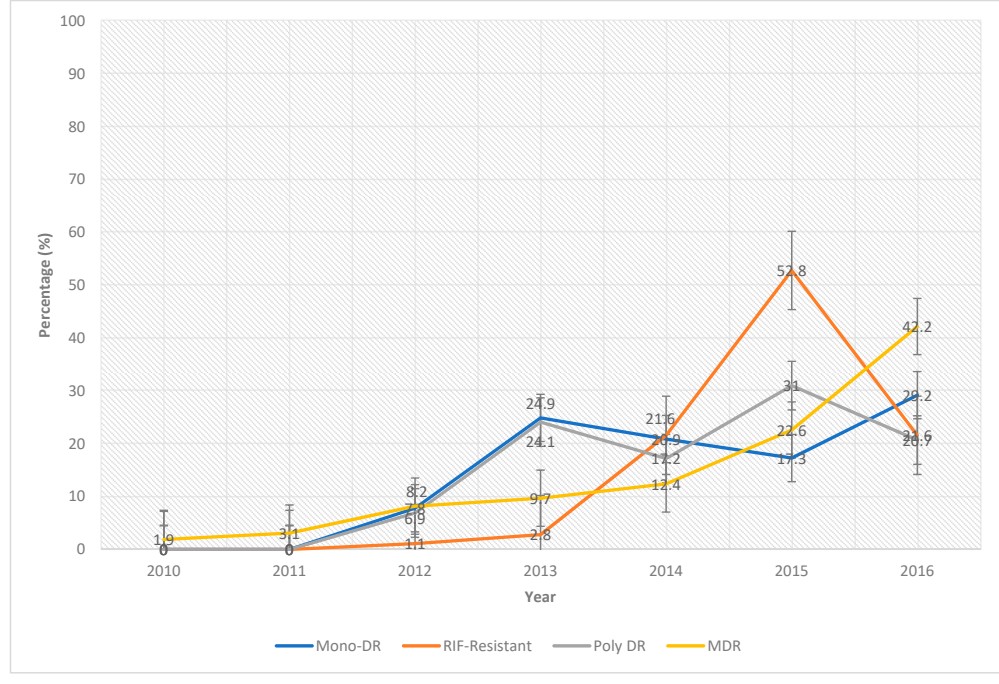

**Figure 1.** Trend of the pattern of presentation of laboratory-confirmed DR-TB cases by year (2010–2016).

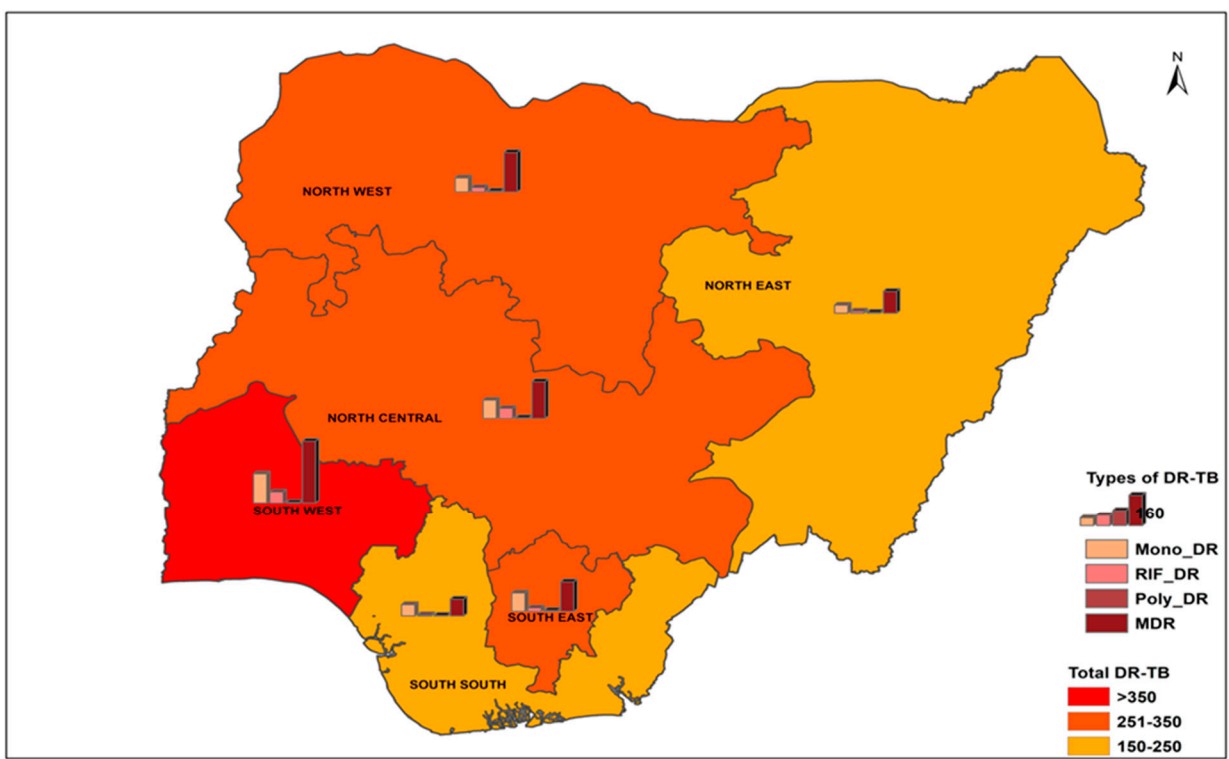

**Figure 2.** Map showing a bar chart of the pattern of presentation by current zone address.

## 4. Discussion

Our study shows that the highest proportion of DR-TB patients (61.4%) had MDR-TB, followed by patients with mono-resistant (27.78%) and rifampicin-resistant (9.28%) TB. DR-TB was highest in the Southwestern zone (36.92%). The number of previous treatments and the current zone address were significantly associated with odds of bacteriologically confirmed DR-TB. The odds of MR-TB, PDR-TB and MDR-TB were significantly higher in the Southwest zone, the odds of PDR-TB and MDR-TB were higher in the North Central zone, while PR-TB odds were significantly higher in the South-south zone. Having received one, two and three previous TB treatments significantly increased the odds of RR-TB, PR-TB and MDR-TB, respectively.

MR-TB, PR-TB, RR-TB and MDR-TB showed an increasing prevalence between 2010 and 2016. RR-TB prevalence was highest in 2015 (52.8%), which coincided with the adoption of Xpert MTB/Rif as the first diagnostic method for all forms of TB in Nigeria. This could result from the improved detection of RR-TB with the use of genexpert. While RR-TB and PR-TB dropped sharply after 2015, MR-TB and MDR-TB continued to rise, after dropping mildly in 2015. This was not unexpected with the expansion of the Line Probe Assay (LPA) and the improvement of DST. Between 2016 and 2017, the number of laboratories capable of LPA evaluation of first-line anti-TB drugs increased from 3 in 2016 to 7 in 2017 [2]. Our findings are, however, different from findings in China, where the number of all DR-TB cases was higher in the period 2009–2010 than in 2011–2015 [18]. This may be because 2011 coincided with China's development of the new National guideline that scaled up the DOTS program nationally, improving the quality of TB care in public facilities [6,18].

We also found variations across country zones, with significantly higher DR-TB case patterns in Southwest, South-south and North-central Nigeria. Between 2016 and 2017, eight facilities had the capacity to perform culture and DST in Nigeria, with two facilities each in the Southwest, Northwest and South-South, one each in Federal Capital Territory (FCT) and North-central, and none in the North-east and Southeast geo-political zones [2]. One study from Adamawa, a high-TB-burden state in rural Northeast Nigeria, noted earlier that of the 87 Xpert MTB/Rif machines available in Nigeria in 2013, only one was

accessible [19]. The low DR-TB odds in these zones might be largely attributed to poor access to and availability of diagnostic facilities for DR-TB, rather than to a low prevalence. This is in tandem with a report from South Thailand where people in the rural areas would most likely harbor DR-TB due to poor access to culture and DST for first-line drugs [20]. The situation in the Northeast is further compounded by the civil unrest caused by Boko Haram insurgents, resulting in socioeconomic problems, a major factor for the spread of DR-TB [21]. Unlike in our study, residence was not a determinant of MDR-TB in Northern Iran, although a small sample of 22 patients in this study makes this conclusion unreliable [22]. Therefore, understanding these zone-specific DR-TB patterns in Nigeria will help clinicians and other frontline health workers in tuberculosis programs to determine the best treatment options and combination approaches for DR-TB treatment.

Similar to our study that found 61% of MDR-TB, studies from Bangalore, India and Turkey reported high proportions of DR-TB [23–25]. However, our result was lower than the DR-TB proportion of 72.4% recorded in Bangalore and that of 76.4% indicated for Nigeria [23,26], but higher than those calculated in India (20.4%), Turkey (38.7%) and Northern Iran (8.2%) [22,24,25] and comparable to that of 69.4% found in Northern Kernataka [27]. The proportion of MR-TB in our study was lower than that reported for Turkey (37.8%) but higher than that (4.2%) for Bangalore [23,25]; however, the reported percentage of 9.1% for RR-TB in our study is comparable to those of 10.8% recorded in Northern Thailand and 10.2% determined in China [20,28]. According to the WHO, a rifampicin resistance of less than 3% for non-MDR-TB is a good quality performance indicator [29]. A German study reported that 13.5% of resistant TB was not resistant to rifampicin, and 80% of patients with rifampicin resistance without isoniazid resistance were mono-resistant [12]. A previous study from South Ethiopia also found 1.4% of cases with non-MDR rifampicin-resistant TB, which supports the use of rifampicin resistance as a surrogate marker for MDR-TB [30]. We could not make this conclusion in our study, since it lacked disaggregation to determine if the RR-TB prevalence was specifically associated with MDR-TB or non-MDR TB.

Similar to our study, patients who received more than two previous anti-TB treatments (ATT) (compared to one) had a 2.4 times higher risk of DR-TB in India and 3.25 times increased odds of confirmed MDR-TB in Mali [31,32], while a study from Ethiopia found an association between two or more previous TB treatments and any type of drug resistance [30]. In Bangladesh, 54% of the 293 TB patients who had received TB treatment more than once were MDR-TB, and 97% MDR-TB patients in the Philippines reported a previously failed Category 2 treatment, indicating they were treated at least twice for TB in the past [33,34]. This finding could be related to the multiple exposure of the mycobacterium to anti-TB medications that increased with the number of past TB treatment episodes. This led to the development of adaptive mechanisms that resulted in resistance. A study found that the risk of DR-TB increased with the length of exposure to ATT, and patients previously treated for <6 months were less likely to develop DR-TB compared with those treated for >6 months [18,35]. Therefore, the number of previous TB treatments can be used as an imperfect proxy for increased duration and exposure to first-line anti-TB, increasing the risk of MDR-TB. In addition to documenting the history of previous TB treatments, healthcare workers working in TB programs need to probe for the number of previous treatments, ask for medication packs and possibly check regimen nomenclatures. This could be a good strategy to assess the specific risk of DR-TB patterns among previously treated patients. It is also important that all TB patients who were treated in the past undergo DST at the start of the TB therapy to improve their outcome.

## 5. Strengths and Weaknesses

The strengths of this study include the representativeness of the data and the large sample of data collected across the six geopolitical zones in Nigeria. In addition, the number of previous treatment episodes, which commonly is not evaluated, was considered. However, there are some limitations. Information on alcohol, smoking, diabetes,

unemployment, drug abuse is not routinely reported at the national level; hence, it was not included in the data collected for routine risk factor analysis. Further, we could not make any casual conclusion because of the cross-sectional nature of the study. In addition, we did not differentiate non-MDR rifampicin resistance from MDR-TB resistance, which could have helped to determine the prevalence of rifampicin mono-resistance and the credibility of using rifampicin resistance as a surrogate marker for MDR-TB in Nigeria. However, one study with a small sample size of 79 participants in Northeastern Nigeria found that all the four RR-TB cases were culture-confirmed MDR-TB [19]. Therefore, it is possible that RR-TB accurately predicts MDR-TB in this zone, but it is difficult to make such conclusion for all zones in the country. Studies from Peru and California noted earlier that rifampicin mono-resistance is associated with an increased risk of poor treatment outcome, even when HIV prevalence is low [36,37]. Increased surveillance for rifampicin mono-resistance is needed to determine appropriate treatment regimens. There is also the possibility that our study underestimated DR-TB, since the detection rate of DR-TB in Nigeria is 20%.

## 6. Conclusions

The proportion of MDR-TB, mono-resistant TB and rifampicin-resistant TB in Nigeria was higher than thought. It is pivotal to expand the provision of DST and culture, providing greater diagnostic capacity for the different types of resistant TB in Nigeria. As DR-TB patterns varied across zones in Nigeria, it is important that clinicians and other frontline healthcare workers consider zone-specific DR-TB patterns to determine the best treatment options and regimen combinations for DR-TB management in each zone with the aim to optimize clinical outcomes.

Taking into account the number of previous TB treatment episodes is recommended as a good strategy to predict the risk of various DR-TB patterns in these populations. This may involve asking patients or checking their prescriptions of TB medications in prior treatments. The finding of high rifampicin resistance (though not disaggregated as mono-resistance) needs to be further investigated in future studies. Rifampicin is one of the two most useful drugs for TB treatment, and resistance to this drug can lead to the failure of TB programs. NTBLCP experts need to weigh in on this as they plan to introduce chemoprophylaxis for the management of latent tuberculosis, some of which contain rifampicin.

**Author Contributions:** Conceptualization, O.O., K.E.O. and Y.O.; methodology, K.E.O., B.P.A. and V.A.A.; software, O.O. and Y.O.; validation, O.O. and Y.O.; formal analysis, O.O.; investigation, O.O.; resources, O.O.; data curation, Y.O.; writing—original draft preparation, O.O. and V.A.A.; writing—review and editing, K.E.O., Y.O. and B.A.O.; visualization, B.P.A.; project administration, O.O.; funding acquisition, O.O. All authors have read and agreed to the published version of the manuscript.

**Funding:** Research reported in this publication was supported by the Fogarty International Center and the National Institutes of Mental Health of the National Institutes of Health under grant number D43 TW010543. The content is the sole responsibility of the authors and does not necessarily represent the official views of the National Institutes of Health and the National TB and Leprosy Control Program (NTBLCP) of the Federal Ministry of Health, Nigeria.

**Institutional Review Board Statement:** The study was conducted in accordance with the Declaration of Helsinki and approved by the National Health Research Ethics Committee of Nigeria (NHREC/01/01/2007), Jos University Teaching Hospital Ethics Committee (JUTH/DCS/ADM/127/XXIX/1586) and the Oyo State Research Ethics Review Committee (13/479/1370 AD). The study also met the Boston University Institutional Review Boards' waiver criteria for analysis of routinely collected program data (H-38912). Patient information was anonymized and de-identified prior to analysis. Since the program data was routinely collected, the designated ethics committees approved the study and waived consent.

**Informed Consent Statement:** Not applicable.

**Data Availability Statement:** The datasets generated and analyzed during the current study are not publicly available. Data are however available from the authors upon reasonable request and with permission of the National Tuberculosis, Leprosy and Buruli ulcer Control Program (NTBLCP).

**Acknowledgments:** The authors would like to thank Victor Babawale, Adebola Lawason and Joseph Kuye. We dedicate this paper to the loving memory of late Ayodele Awe and late Lovett Lawson. Oladimeji is a Visiting Researcher at the Department of Global Health and Population, Harvard T.H. Chan School of Public Health (HSPH). He is grateful for the platform HSPH has given him for his career growth.

**Conflicts of Interest:** The authors declare no conflict of interest.

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
