# Peer review of "Patterns of Presentation of Drug-Resistant Tuberculosis in Nigeria: A Retrospective File Review"

_2036-7481, doi:10.3390/microbiolres13030043_

Round 1
Reviewer 1 Report
The manuscript recruited drug resistant TB patients between 2010 and 2016 in Nigeria and investigated the correlation of TB drug resistance with socioi-demographic factors including age, sex, treatment, drug resistance, areas, and HIV status.
There are some concerns about this manus.
1. The writing quality is poor. There are lots of grammar problems and unstandardized writing. The authors need to improve it. Here listed parts of them.
“DR-TB” in line30 is the abbreviation of drug resistant tuberculosis, which is the first appearance. According to the academic writing, it should be written as full name followed with the abbreviation.
In line177, “…participants with…” should be “…participants including…”.
In line183, “…of the respondents had multi-drug resistancefollowed…” should have a space between the word resistance and the word followed.
2. Line61 to line62 introduced new infected and death data in 2018. Is that the newest data?
3. Table 1 summarized the participants information in this study. Table1 was cited at the end of this paragraph (line181). It is better to cite Table 1 at the beginning, such as “As shown in Table 1”.
4. Line177 to line186 did not describe the information from Table1 clearly. There are some grammar problems. The authors should clearly describe age distribution. In table1, there are 2476 samples from patient group, including 567 newly treated and 1909 previously treated. In total there are 2555 participants. Are the other 79 samples from untreated ones? The authors should reflect that information in Table1. 1909 samples are from previously treated patient. Why are only 823 samples shown in “No of previous treatment”? In addition, 1303 samples showed HIV status, negative or positive. Did the other samples have no HIV information? The authors need to modify Table 1 and reorganize the related text.
5. How did you calculate the statistic significance? The authors should add those detailed information.
Author Response
June 21st, 2022
Manuscript ID: microbiolres-1768798
Pattern of presentation of drug-resistant Tuberculosis in Nigeria; a retrospective file review.
Point-by-Point response to the reviewer comments
Dear Editor,
We thank the handling editor and reviewers for reviewing our manuscript. We are pleased that there is interest in this topic and are grateful for the constructive comments that have improved the manuscript.
We have addressed each of the reviewers’ comments in a point-by-point manner in the table below. The reviewers’ comments are on the left column, and our responses are highlighted in blue font on the right. We have also made corresponding changes in the manuscript, indicating the revised sections with "track changes. While hoping that these changes will meet with your favorable consideration, we hold ourselves at your disposition for any further clarifications.
|
Editorial requests |
|
|
|
|
|
Reviewer 1 |
|
|
The writing quality is poor. There are lots of grammar problems and unstandardized writing. The authors need to improve it. Here listed parts of them |
This is noted and has been addressed. |
|
“DR-TB” in line30 is the abbreviation of drug resistant tuberculosis, which is the first appearance. According to the academic writing, it should be written as full name followed with the abbreviation |
“DR-TB” has been written in full with the abbreviation in parenthesis |
|
In line177, “…participants with…” should be “…participants including…”. |
The correction has been made accordingly |
|
In line183, “…of the respondents had multi-drug resistancefollowed…” should have a space between the word resistance and the word followed |
A space has been inserted between the word “resistance” and “followed”. |
|
Line61 to line62 introduced new infected and death data in 2018. Is that the newest data? |
This is the data available |
|
Table 1 summarized the participants information in this study. Table1 was cited at the end of this paragraph (line181). It is better to cite Table 1 at the beginning, such as “As shown in Table 1” |
Table 1 has been cited at the beginning of the sentence. |
|
Line177 to line186 did not describe the information from Table1 clearly. There are some grammar problems. The authors should clearly describe age distribution. In table1, there are 2476 samples from patient group, including 567 newly treated and 1909 previously treated. In total there are 2555 participants. Are the other 79 samples from untreated ones? The authors should reflect that information in Table1. 1909 samples are from previously treated patient. Why are only 823 samples shown in “No of previous treatment”? In addition, 1303 samples showed HIV status, negative or positive. Did the other samples have no HIV information? The authors need to modify Table 1 and reorganize the related text |
Lines 177 to 186 have now been addressed
This is missing record issues and the annotation for missing record has been included |
|
How did you calculate the statistic significance? The authors should add those detailed information |
This is described in the Data analysis section |
Sincerely,
Regards,
--------------------------
Olanrewaju Oladimeji II MB; BS, MSc, MPA, FRSPH, PhD
Research Chair: Department of Public Health, Walter Sisulu University, Eastern Cape, South Africa
Visiting Scholar: Harvard T.H. Chan School of Public Health, Boston, USA
Adjunct Professor: Faculty of Health Sciences, Durban University of Technology, South Africa.
email: droladfb@gmail.comIIooladimeji@wsu.ac.za

Reviewer 2 Report
Comments for authors:
1. Can authors include the types of specimens that is used for detection of TB by GeneXpert method?
2. Authors should includes more recent (2018 to 2022) references in this study.
Author Response
June 21st, 2022
Manuscript ID: microbiolres-1768798
Pattern of presentation of drug-resistant Tuberculosis in Nigeria; a retrospective file review.
Point-by-Point response to the reviewer comments
Dear Editor,
We thank the handling editor and reviewers for reviewing our manuscript. We are pleased that there is interest in this topic and are grateful for the constructive comments that have improved the manuscript.
We have addressed each of the reviewers’ comments in a point-by-point manner in the table below. The reviewers’ comments are on the left column, and our responses are highlighted in blue font on the right. We have also made corresponding changes in the manuscript, indicating the revised sections with "track changes. While hoping that these changes will meet with your favorable consideration, we hold ourselves at your disposition for any further clarifications.
|
Reviewer 2 |
|
|
Can authors include the types of specimens that is used for detection of TB by GeneXpert method? |
This has been included in the first sentence under “DRTB case finding and diagnostic coverage” section. |
|
Authors should include more recent (2018 to 2022) references in this study |
We struggled to find new information on the study given the changes that have been brought in by WHO. |
Sincerely,
Regards,
--------------------------
Olanrewaju Oladimeji II MB; BS, MSc, MPA, FRSPH, PhD
Research Chair: Department of Public Health, Walter Sisulu University, Eastern Cape, South Africa
Visiting Scholar: Harvard T.H. Chan School of Public Health, Boston, USA
Adjunct Professor: Faculty of Health Sciences, Durban University of Technology, South Africa.
email: droladfb@gmail.comIIooladimeji@wsu.ac.za

Round 2
Reviewer 1 Report
The authors answered the comments and the writing improved a lot. But to publish, I strongly recommend to ask the professional person further improve the writing. There are still some issues. Here listed part of them.
1. The font size of "drug-resistant tuberculosis" is smaller than others (line28).
2. There are grammar problems in line44-46.
Author Response
Pattern of presentation of drug-resistant Tuberculosis in Nigeria; a retrospective file review
Point-by-Point response to the reviewer comments
Dear Editor,
We thank the Handling Editor and reviewers for their efforts. We are pleased that there is interest in this topic and are grateful for the constructive comments that have improved the manuscript.
We have addressed each of the reviewers’ comments in a point-by-point manner in the table below. The reviewers’ comments are on the left column, and our responses are highlighted in blue font on the right. We have also made corresponding changes in the manuscript, indicating the revised sections with "track changes. While hoping that these changes will meet with your favorable consideration, we hold ourselves at your disposition for any further clarifications.
|
S/N |
Comments |
Response |
|
Reviewer 2 |
||
|
1 |
The authors answered the comments and the writing improved a lot. But to publish, I strongly recommend to ask the professional person further improve the writing. There are still some issues. Here listed part of them. |
All the comments as listed have now been thoroughly addressed in the manuscript. Professional editing certificate is now provided |
|
2 |
The font size of "drug-resistant tuberculosis" is smaller than others (line28). |
This font size has now been changed to “Segoe UI Historic 12” on line 29 |
|
3 |
There are grammar problems in line44-46. |
The grammar problems have now been attended to. This section has been rephrased |
Yours’
Olanrewaju Oladimeji, MB;BS, MSc, MPA, MACE, FRSPH, PhD, Cert. Clinical Trials (Harvard), Postdoc (Harvard)
Epidemiologist and Research Chair: Department of Public Health, Walter Sisulu University, Eastern Cape, South Africa
Visiting Scholar: Harvard T.H. Chan School of Public Health, Boston, USA
Adjunct Professor: Public Health, College of Health Science, Bowen University, Iwo, Nigeria
Adjunct Professor: Faculty of Health Sciences, Durban University of Technology, Durban, South Africa
Visiting Professor: Department of Community Medicine, College of Health Sciences, University of Jos, Nigeria
Editorial Board Member: Nature Scientific Reports
Associate Editor: BiomedCentral Public Health; WC1X 8HB, United Kingdom
Academic Editor: PLOS ONE Journal; 1160 Battery Street, Suite 225, San Francisco, CA 94111
Emails: ooladimeji@wsu.ac.za II oladimeji@hsph.harvard.edu II droladfb@gmail.com Cell: +27622583986
www.sayas.org.za/?team=olanrewaju-oladimeji
www.researchgate.net/profile/Olanrewaju-Oladimeji
